# Inflammation-Associated Carcinogenesis in Inflammatory Bowel Disease: Clinical Features and Molecular Mechanisms

**DOI:** 10.3390/cells14080567

**Published:** 2025-04-09

**Authors:** Tadakazu Hisamatsu, Jun Miyoshi, Noriaki Oguri, Hiromu Morikubo, Daisuke Saito, Akimasa Hayashi, Teppei Omori, Minoru Matsuura

**Affiliations:** 1Department of Gastroenterology and Hepatology, Kyorin University School of Medicine, Tokyo 181-8611, Japan; jmiyoshi@ks.kyorin-u.ac.jp (J.M.); n_oguri@ks.kyorin-u.ac.jp (N.O.); hiromu-morikubo@ks.kyorin-u.ac.jp (H.M.); straw@zb3.so-net.ne.jp (D.S.); teppei-omori@ks.kyorin-u.ac.jp (T.O.); mmatsuura@ks.kyorin-u.ac.jp (M.M.); 2Department of Gastroenterology and Hepatology, Kyorin University Suginami Hospital, Tokyo 166-0012, Japan; 3Department of Pathology, Kyorin University School of Medicine, Tokyo181-8611, Japan; hayashia@ks.kyorin-u.ac.jp

**Keywords:** dysplasia, inflammatory bowel disease, ulcerative colitis-associated neoplasia (UCAN), Crohn’s disease, p53

## Abstract

Inflammatory bowel disease (IBD), comprising ulcerative colitis (UC) and Crohn’s disease (CD), is a chronic condition marked by persistent intestinal inflammation of unknown etiology. Disease onset involves genetic predisposition and environmental factors that disrupt the intestinal immune homeostasis. The intestinal microbiome and immune response play pivotal roles in disease progression. Advances in molecular therapies and early interventions have reduced surgery rates; however, colorectal cancer (CRC) remains a significant concern, driven by chronic inflammation. In UC, the risk of UC-associated neoplasia (UCAN) increases with disease duration, while CD patients face elevated risks of small intestine, anal fistula, and anal canal cancers. Endoscopic surveillance is advised for UCAN, but optimal screening intervals remain undefined, and no established guidelines exist for CD-associated cancers. UCAN morphology often complicates detection due to its flat, inflammation-blended appearance, which differs pathologically from sporadic CRC (sCRC). UCAN is frequently surrounded by dysplasia, with p53 mutations evident at the dysplasia stage. IBD-associated gastrointestinal cancers exemplify inflammation-driven carcinogenesis with distinct molecular mechanisms from the adenoma-carcinoma sequence. This review explores the epidemiology, risk factors, clinical and pathological features, current surveillance practices, and molecular pathways underlying inflammation-associated cancers in IBD.

## 1. Introduction

Inflammatory bowel disease (IBD) is a chronic inflammatory disorder characterized by repeating relapse and remission that includes ulcerative colitis (UC) and Crohn’s disease (CD) as two major disease phenotypes. The pathophysiology of IBD remains unestablished, and today’s treatment goals are the induction and maintenance of remission. With the advancement of therapeutic options and strategies, the long-term prognosis of IBD has improved [1,2]. However, the carcinogenesis of colorectal cancer (CRC) due to chronic inflammation remains a significant clinical challenge.

This review provides a comprehensive overview of inflammation-associated CRC in IBD, covering clinical perspectives such as epidemiology, risk factors, endoscopic and pathological features, and endoscopic surveillance, as well as basic research exploring the distinct carcinogenic mechanisms that differ from those of sporadic CRC.

## 2. Epidemiology of Inflammation-Associated Neoplasia in Inflammatory Bowel Disease (IBD)

### 2.1. UC-Associated Colorectal Neoplasia (UCAN)

UC, the risk of inflammation-associated carcinogenesis in the colonic mucosa, increases with the disease’s duration. In 2001, a meta-analysis by Eaden et al. [3] revealed a cumulative incidence of CRC in patients with UCs of 1.6% at 10 years after disease onset, 8.3% at 20 years, and 18.4% at 30 years.

On the other hand, recent studies from referral centers based on long-term surveillance programs targeting patients with UC indicate that the cumulative incidence of UC-associated CRC is not as high as previously reported. According to a study by Choi et al. [4], the cumulative incidence of CRC at 10, 20, and 30 years after UC onset was 0.07%, 2.90%, and 6.70%, respectively. Similarly, a report by Kishikawa et al. [5] showed the cumulative incidences of CRC to be 0.7% after 10 years, 3.2% after 20 years, and 5.2% after 30 years, which are comparable. However, the cumulative incidences of dysplasia, a precursor lesion of CRC, remained high at 3.3%, 12.1%, and 21.8% over 10, 20, and 30 years, respectively, necessitating careful monitoring [5]. A meta-analysis by Lutgens et al. [6] in 2013 revealed that the CRC risk (pooled standardized incidence ratio [SIR]) in patients with UC was significantly higher than that in the general population across both referral center cohorts and population-based cohorts, with marked variations depending on the study population (referral center cohort: SIR = 8.3, 95% confidence interval [CI] = 5.9–10.7; population-based cohort: SIR = 1.7, 95% CI = 1.03–2.4). Another meta-analysis by Jess et al. [7], which synthesized data from eight population-based cohort studies, showed that the risk of CRC in patients with UC was 2.4 times higher than that in the general population. Furthermore, a population-based cohort study comparing 96,447 patients with UC and 949,207 individuals from the general population in Sweden and Denmark showed that the incidence and mortality risk of CRC (hazard ratio [HR]) were significantly higher in the UC group (incident CRC: HR = 1.66, 95% CI = 1.57–1.76; CRC mortality: HR = 1.59, 95% CI = 1.46–1.72), although both risks declined over time [8]. Additionally, the risk of carcinogenesis is reported to increase with the extent of colonic involvement in UC [9]. Overall, while the cumulative incidence of UCAN may not be as high as previously reported, careful monitoring for CRC remains necessary for patients with UC.

### 2.2. Crohn’s Disease (CD)-Associated Gastrointestinal Neoplasms

Similar to UC, CD is associated with an increased risk of gastrointestinal neoplasms as compared with the general population. In the first meta-analysis on CD-related gastrointestinal neoplasms by Jess et al. [10], it was reported that the risks of CRC and small bowel cancer in patients with CD were significantly higher than those in the general population (CRC: SIR = 1.9, 95% CI = 1.4–2.5; small bowel cancer: SIR = 27.1, 95% CI = 14.9–49.2). A subsequent meta-analysis by Canavan et al. [11], which included 12 population-based cohort studies, yielded similar results (CRC: SIR = 2.5, 95% CI = 1.3–4.7; small bowel cancer: SIR = 31.2, 95% CI = 15.9–60.9). The cumulative incidence of CRC in patients with CD was reported as 2.9% at 10 years, 5.6% at 20 years, and 8.3% at 30 years [11]. The risk of CRC is significantly increased in patients with colonic involvement, whereas those with small bowel-only CD have a risk comparable to that of the general population [6,11]. Additionally, the CRC risk involving the colon in patients with CD is nearly equivalent to that in patients with UC [6,11]. An umbrella review that reanalyzed 24 meta-analyses confirmed that the overall risk of gastrointestinal neoplasms in patients with CD was significantly higher than that in the general population (relative risk [RR] = 1.56, 95% CI = 1.10–2.23) [12]. By location, the risks were particularly elevated in the small intestine (RR = 11.9), colon (RR = 2.30), rectum (RR = 1.85), and anus (RR = 4.52). By contrast, no significant increase in cancer risk was observed in other regions, such as the upper gastrointestinal tract, pharynx, esophagus, or stomach [12]. Despite the high relative risk of small bowel cancer in patients with CD, its proportion among all gastrointestinal neoplasms is only about 2% [13], suggesting that the absolute risk remains low [11,14,15]. Regarding common sites of CD-associated CRC, studies in Western countries have reported a predilection for the right colon [16,17], whereas in Japan, the rectum and anal canal are more frequently affected [18]. However, a meta-analysis by Uchino et al. [19] demonstrated that 63.1% of CD-associated CRC cases in Western reports occurred in the left colon. Western guidelines also emphasize the need for vigilance regarding cancer risk in the anorectal region [15]. Most CD-associated small bowel cancers arise from inflamed areas of the small intestine, primarily the ileum. They exhibit distinct characteristics as compared with sporadic small-bowel cancers, which can arise from anywhere in the small intestine, including the duodenum [13,15]. Overall, patients with CD are considered to have higher risks of gastrointestinal neoplasms as compared with the general population. Inflamed lesions, not only in the colon but also in the anorectal area and small intestine, can increase the risk of developing cancer.

## 3. Risk Factors for Inflammation-Associated Carcinogenesis in IBD

Clinical risk factors for IBD-related cancer have been described in various guidelines [20,21]. These risk factors include patient-related and treatment-related factors. This section discusses the clinical risk factors for inflammation-associated carcinogenesis.

### 3.1. Patient-Related Risk Factors

Known risk factors for IBD-related CRC include being male [22] and having a family history of CRC [23]. Although family history is a recognized risk factor for sporadic CRC (sCRC) in the non-IBD population, an increased risk of developing IBD-related CRC has also been reported when a close relative, particularly a first-degree relative, has developed CRC [24]. On the other hand, no clear association has been established for race, smoking, or a family history of IBD [22,25].

As a comorbidity, primary sclerosing cholangitis (PSC) is often associated with IBD, and its presence is known to markedly increase the risk of CRC in patients with IBD [26]. Therefore, regular endoscopic surveillance is strongly recommended for patients with PSC-complicated IBD [27]. Furthermore, various factors related to IBD, such as disease phenotype and activity, are associated with CRC risk. First, there is no difference in CRC risk between IBD types (UC or CD) [21]. However, the risk of small intestinal cancer is significantly elevated in patients with CD [24,25,28,29]. Second, the age at IBD onset is believed to be associated with subsequent CRC risk, with particularly high risks in patients who develop young-onset IBD (<30 years of age) [25]. This is thought to be due to relatively high disease activity and the difficulty in controlling inflammation, which leads to persistent histological inflammation.

The incidence of CRC is known to increase when IBD has been present for more than 8 to 10 years [3]. However, UC-associated CRC can occur even within the first 8 years [30], and the possibility of its development should be considered even in patients with shorter disease durations. On the other hand, for small bowel cancer in CD, an increased risk has been reported with increasing age at onset [31].

The extent and severity of inflammation are also associated with carcinogenesis. In UC, the risk of CRC is comparable to that of the general population in patients with the proctitis type [22], while those with more extensive inflammation, such as left-sided colitis or pancolitis, have a particularly high risk of CRC [25]. This also applies to patients with CD, among whom those with more than 50% colorectal involvement are at increased risk of CRC [25].

The severity of inflammation is also closely associated with the risk of carcinogenesis. For example, patients with repeatedly high levels of endoscopic inflammation have a greater risk of CRC than those who are maintained in remission [26,31]. Additionally, the Geboes score, a histological grading system for UC, has been linked to carcinogenesis [32].

In small bowel cancer, CD with a stricturing phenotype is associated with an increased risk of carcinogenesis [29,33]. Fistula-associated adenocarcinoma and anal-fistula cancer may also develop at fistula-forming sites and in patients with CD involving anal lesions, respectively [33]. Patients with these complications require careful monitoring, and determining the appropriate indication for surgery is crucial.

### 3.2. Treatment-Related Risk Factors

Although the long-term use of immunosuppressants, such as thiopurines and Janus kinase inhibitors, is considered to be associated with carcinogenesis, there is no solid evidence confirming this relationship. In particular, thiopurines are known to increase the risk of non-intestinal cancers, such as non-melanoma skin cancer and lymphoma, leading to concerns about their safety profiles [34]. However, avoiding thiopurine use entirely is not recommended; appropriate use may help reduce inflammation and indirectly lower cancer risk. The European Crohn’s and Colitis Organisation guidelines also state that there is insufficient evidence to support either recommending or avoiding thiopurines [27].

The 5-aminosalicylic acid has been suggested to have a protective effect against CRC because of its anti-inflammatory and antioxidant properties [35]. Although various meta-analyses have been conducted, the effect remains uncertain [25]. No solid evidence is currently available regarding the impact of biologics, such as anti-tumor necrosis factor-alpha (anti-TNF-α) inhibitors and interleukin (IL)-12/23 inhibitors, on the risk of carcinogenesis [34]. When using biologics, treatment strategies must carefully balance inflammation control with the potential risk of carcinogenesis.

## 4. Endoscopic and Pathological Features of UCAN

### 4.1. Endoscopic Features of UCAN

Colonoscopy is the most accurate imaging modality for diagnosing UCAN, similar to diagnosing conventional epithelial tumors. Today, several major guidelines recommend an endoscopic approach for surveillance for UCAN [21,27,36]. However, unlike typical epithelial tumors, UCAN develops against a background of chronic inflammation and often has a proliferative zone located in the middle to deep layers of the mucosa, with differentiation tending toward the surface. As a result, diagnosing early UCAN lesions based solely on surface appearance is challenging. Given the complexity of UCAN diagnosis, a stepwise approach using multiple endoscopic modalities is required, including white-light endoscopy, chromoendoscopy, and magnified image-enhanced endoscopy (IEE).

#### 4.1.1. Diagnosis Using White-Light Endoscopy and Chromoendoscopy

UCAN exhibits various morphological types, including polypoid, superficial, and mixed (unclassifiable) forms (Figure 1A–G). The Surveillance for Colorectal Endoscopic Neoplasia Detection and Management in Inflammatory Bowel Disease Patients: International Consensus Recommendations (SCENIC) guidelines propose a macroscopic classification system that categorizes lesions as either polypoid dysplasia (pedunculated or sessile) or nonpolypoid dysplasia (superficial elevated, flat, or depressed), with additional descriptors for ulceration and lesion margin clarity [37]. Polypoid lesions, such as pedunculated or subpedunculated types, are rare in UCAN. More commonly, lesions exhibit granular elevations with indistinct margins and a villous surface structure [38]. High-grade dysplasia with a flat or depressed morphology often appears as a well-demarcated erythematous area [39]. Careful observation of surface structure and color changes after indigo carmine staining enhances the visibility of the lesion’s macroscopic features and boundaries, making chromoendoscopy a recommended adjunctive diagnostic method [38].

#### 4.1.2. Diagnosis Using Magnifying Endoscopy and IEE

Observation of microvascular and surface structures using IEE can sometimes improve the clarity of lesion margins as compared with white-light imaging [40]. In the diagnosis of conventional adenomas and carcinomas, magnifying endoscopy is an essential tool. However, while various studies have investigated its utility for UCAN, evidence remains insufficient to establish a clear correlation between abnormalities in microvascular and surface structures and the degree of dysplasia in UCAN [41,42,43]. The usefulness of ultra-magnifying endoscopy has recently been reported, and its future applications are highly anticipated [44].

### 4.2. Pathological Features of UCAN

The World Health Organization classification designates intraepithelial tumors arising in the background of IBD as “inflammatory bowel disease-associated dysplasia of the colorectum.” Based on histological morphology, these are categorized into several subtypes, including the intestinal (adenomatous) subtype; serrated subtype; mucinous type, a subtype with eosinophilic cytoplasm and marked goblet cell depletion; crypt cell subtype; and mixed type, in which multiple histological features coexist [45]. Given the diverse histological appearances of dysplasia, differentiation from reactive atypia is often challenging. However, distinct atypical epithelial changes, such as dystrophic goblet cells, endocrine cell hyperplasia, and Paneth cell metaplasia, serve as valuable morphological criteria. Additionally, dysplasia is known to exhibit TP53 genetic abnormalities at an early stage of tumor development, making abnormal p53 protein expression a crucial immunohistochemical marker for distinguishing dysplasia from reactive atypia and sporadic adenomas. Unlike conventional colorectal adenomas and carcinomas, in which p53 overexpression correlates with the degree of atypia, dysplasia frequently exhibits p53 overexpression even at the low-grade stage (Figure 2A–E) [46]. Consequently, when a low-grade lesion demonstrates a p53 overexpression pattern, the likelihood of dysplasia is high, making it a significant diagnostic feature. Moreover, in low-grade adenoma (low-grade dysplasia) with surface differentiation, p53 overexpression, if present, often diminishes in the superficial layer, whereas high-grade adenoma (high-grade dysplasia) typically exhibits diffuse p53 overexpression across the entire epithelial layer. Ki-67 staining patterns further aid differentiation: dysplasia typically exhibits a bottom-up proliferation pattern, where the proliferative zone is located in the deeper to middle layers of the crypts, in contrast to sporadic adenomas, which display a top-down proliferation pattern with the proliferative zone extending from the superficial to middle layers (Figure 3) [47]. These immunohistochemical staining patterns are critical in distinguishing dysplasia from reactive atypia and sporadic adenomas. The Riddell system is widely used for biopsy diagnosis of dysplasia [48].

## 5. Surveillance Colonoscopy: Current Status and Challenges

Repeated relapses, remissions, and persistent inflammation in UC increase the risk of developing UCAN. A 2001 report estimated the cumulative incidence of UCAN to be approximately 20% at 30 years [3]. However, more recent reports from Japan indicate a lower cumulative incidence: 0.7% at 10 years, 3.2% at 20 years, and 5.2% at 30 years [5]. Despite this decline, UCAN tends to have an earlier onset than sCRC and a higher proportion of poorly differentiated and mucinous carcinomas [49,50], which may contribute to a poorer prognosis. Therefore, surveillance remains critically important.

### 5.1. Targets for Surveillance Colonoscopy

Most guidelines recommend initiating surveillance in cases of total or left-sided colitis (excluding proctitis) 8 years after diagnosis or the onset of UC symptoms [21,27,33,36]. However, it is important to note that UCAN can develop within a shorter timeframe, particularly when UC is diagnosed after a prolonged period of illness. While surveillance intervals vary between Japan, Europe, and the United States, they generally range from 1 to 3 years, with the interval determined based on underlying risk factors.

Risk factors for UCAN development include the disease duration (long-term cases, younger patients), extent of disease (total colitis > left-sided colitis > proctitis), family history of CRC, disease type (chronic persistent > relapsing-remitting), and the presence of PSC [51].

### 5.2. Timing and Methods of Surveillance Colonoscopy

Regarding the timing of surveillance, diagnosing lesions is challenging when intestinal mucosa inflammation is present. Therefore, as a general rule, surveillance should be conducted during remission. While a Mayo Endoscopic Subscore of 0 is ideal, surveillance is recommended when the mucosa has a Mayo Endoscopic Subscore of ≤1. In the presence of mucosal inflammation, distinguishing between inflammatory cellular atypia and tumors can be difficult both endoscopically and pathologically.

Random biopsy, involving four biopsies every 10 cm (minimum of 33 biopsies), has been recommended as a surveillance method. However, a randomized controlled trial conducted in Japan compared the detection rates of tumor lesions using random biopsy and step biopsy, revealing that both methods had similar detection rates, while the step biopsy group required significantly fewer biopsies [52]. Therefore, step biopsy is recommended for patients with stable inflammation.

On the other hand, studies have reported that random biopsy increases the tumor detection rate by 15% [53] and that certain lesions may be difficult to detect without it [30]. Additionally, because patients with persistent inflammation are at high risk for UCAN, random biopsy should be considered when necessary.

When performing a step biopsy, selecting the appropriate mucosal areas for biopsy is crucial. Because more than 80% of UCANs are concentrated in the rectum and sigmoid colon [38], careful observation of the distal colon and rectum is essential.

Western guidelines classify the gross characteristics of UCAN into three morphological types: polypoid lesions, nonpolypoid lesions, and invisible dysplasia. Polypoid lesions are further categorized as pedunculated or sessile, while nonpolypoid lesions are classified into flat elevated, flat, and flat depressed types [37,54]. Studies have shown that flat-type lesions are more common in UCAN than in sporadic tumors, and their boundaries are often indistinct [21,38]. Because of this, it is particularly important to detect flat lesions, which may not be easily identified using conventional white-light observation alone. Effective detection requires paying close attention to subtle erythema and surface irregularities compared with the surrounding mucosa, as well as adjusting the amount of air insufflation during observation.

### 5.3. IEE for Detection of UCAN

To improve lesion detection, the SCENIC consensus statement recommends the use of IEE for diagnosing UCAN [16]. In particular, SCENIC endorses dye-spray chromoendoscopy (DCE) as a surveillance method in standard-definition endoscopy, as reported by Rutter et al. [55].

Dye-spraying is believed to enhance lesion detection by improving localization, defining boundaries, and clarifying gross morphology. Additionally, in high-definition endoscopy, some studies have revealed a higher tumor-lesion detection rate when dyes are used than in cases without dye application [56,57]. However, while high-definition endoscopy with DCE and high-definition endoscopy with random biopsy show no significant difference in tumor detection rates, step biopsies performed with DCE require fewer biopsies [58]. Furthermore, meta-analyses have indicated that the use of DCE with high-definition endoscopy does not necessarily improve lesion detection rates [59,60].

Narrow-band imaging (NBI), texture and color enhancement imaging, and linked color imaging are categorized as virtual IEE methods that do not require dye spraying. A network meta-analysis by Gondal et al. [61] found no significant difference in lesion detection rates among white-light endoscopy, combined DCE, and NBI observation in high-definition endoscopes. However, a recent meta-analysis comparing UCAN detection using virtual IEE, high-definition white-light endoscopy, and dye-enhanced standard-definition endoscopy found them to be comparable [62]. In dysplasia-specific analyses, virtual IEE was inferior to high-definition white-light endoscopy and showed no significant difference from DCE. A multicenter prospective randomized-controlled trial (Navigator Study) compared whole-colon NBI observation using second-generation NBI vs. whole-colon DCE. The study showed that the UCAN detection rate in the whole-colon NBI group was not inferior to that in the whole-colon DCE group, and the examination time was significantly shorter in the NBI group [63]. Additionally, surveillance methods using texture and color enhancement imaging and linked color imaging have only been reported in case studies [40,64]. As a result, the effectiveness of IEE for UCAN detection remains controversial, and further studies are necessary to establish its role.

## 6. Molecular Mechanisms of Colitis-Associated Cancer (CAC) in IBD

Chronic inflammation in IBD increases the risk of developing CAC. Although the precise pathogenesis of CAC remains unclear, carcinogenesis is believed to be driven by a combination of molecular mechanisms, including immune dysregulation, oxidative stress, and genetic alterations [65,66].

### 6.1. Molecular Pathways in CAC

Under chronic inflammation, inflammatory immune cells release excessive inflammatory mediators, including TNF-α, interferon-γ, IL-6, IL-17, IL-23, chemokines, prostaglandins, and matrix metalloproteinases [66,67,68,69,70,71,72,73,74]. These mediators contribute to inflammation-driven carcinogenesis by activating molecular pathways such as Wnt/β-catenin, JAK/STAT, NF-κB, PI3K/AKT, and MAPK/ERK [65,69,75,76]. The Wnt/β-catenin pathway regulates colonic mucosal epithelial-cell differentiation and is commonly activated in sCRC due to *APC* gene mutations, leading to the acquisition of stem cell-like characteristics and cancer development. However, in CAC, Wnt/β-catenin signaling is thought to be activated by inflammatory signals, including NF-κB and TNF-α [75,76]. Additionally, cytokines such as IL-6 and TNF-α, produced during chronic inflammation, promote carcinogenesis by inducing anti-apoptotic proteins through the JAK/STAT pathway [69]. A similar mechanism occurs with NF-κB activation, which inhibits apoptosis and facilitates cancer progression.

### 6.2. Genetic Alterations and Epigenetic Changes in CAC

The inflammatory process induces oxidative stress through the excessive production of reactive oxygen species, leading to DNA damage and oncogenic genetic alterations. These genetic transformations are considered major driving factors in the development of CAC. The genetic and epigenetic alterations involved in CAC carcinogenesis include chromosomal instability (CIN), microsatellite instability (MSI), and the CpG island methylator phenotype, which are also observed in sCRC.

In patients with IBD, CIN is one of the primary mechanisms driving cancer development. In CAC specimens, CIN is present in 85% of cancer sites, 86% of dysplasia sites, and 36% of non-dysplasia sites. CIN leads to the suppression of tumor suppressor genes (e.g., *p53*, *APC*, *IDH1*) and the activation of oncogenic genes (e.g., *K-ras*, *C-myc*, *PIK3CA*) [77,78,79,80]. CIN can also result in aneuploidy, which is strongly associated with carcinogenesis. Aneuploidy is observed in 20–50% of dysplastic lesions and 50–90% of cancers in IBD [81]. Notably, DNA aneuploidy has been detected not only in dysplastic lesions but also in non-dysplastic lesions in UC [82]. Additionally, aneuploidy occurs three times more frequently in the colons of patients with IBD than in healthy individuals [83], suggesting its potential as an early predictive marker for carcinogenesis. MSI, caused by mismatch repair deficiency, is another early feature of CAC. Free radicals produced by neutrophils and macrophages during chronic inflammation can inactivate DNA mismatch repair enzymes, leading to MSI [84]. A study of patients with UC revealed that MSI-H was detected in 67% of CAC cases, 67% of high-grade dysplasia, 33% of low-grade dysplasia, and 25% of inflamed mucosa [85]. Notably, MSI-H was present 2 to 12 years before the final CAC diagnosis [85], highlighting its potential role as an early indicator of malignant transformation.

In the CpG island methylator phenotype, DNA methylation leads to epigenetic instability by inactivating multiple genes. The DNA methylation rate is significantly higher in CAC than in UC and healthy controls [86]. In IBD-associated neoplastic lesions, the *hMLH1* promoter hypermethylation was observed in 46% of specimens with MSI-H, 16% with low-frequency MSI, and 15% with stable MSI [87]. *hMLH1* is a key gene in the DNA mismatch repair pathway, which maintains genomic stability by correcting replication errors [88]. In UC, the CpG methylation density increases in dysplasia and carcinoma as compared with normal epithelial cells. Additionally, DNA hypermethylation of CpG islands in gene promoter regions is thought to occur before the morphological onset of tumor formation [89,90]. This phenomenon is likely driven by increased cell turnover and excessive oxidative stress in UC.

### 6.3. Genetic Drivers of CAC

Comprehensive gene sequencing analysis has revealed that key driver genes for cancer development, such as *APC*, *K-ras*, *TP53*, *PIK3CA*, *SMAD4*, and *MYC*, are common to both CAC and sCRC [78,79,91,92,93]. However, the carcinogenic mechanisms in CAC differ from the classical adenoma–carcinoma sequence observed in sCRC and are instead characterized by an inflammation–dysplasia–carcinoma sequence [78,79,91,92,93] (Figure 4). While various gene mutations have been reported, one major observation is that *TP53* mutations occur at the dysplasia and early cancer stages in CAC [78,79,92,93]. By contrast, *APC* and *K-ras* mutations in CAC tend to occur later in disease progression and are observed at a lower frequency [33,78,79,92,93]. This contrasts with sCRC, in which the adenoma–carcinoma sequence follows a pattern where *APC* mutations appear first, followed by *K-ras* mutations, with *TP53* mutations occurring at a later stage [91,94]. Furthermore, Yaeger et al. [79] identified potential therapeutic target mutations in an analysis of 47 CAC cases, including *IDH1* R132 mutations; *FGFR1*, *FGFR2*, and *ERBB2* amplifications; *BRAF* V600E mutations; and EML4-ALK fusion proteins. Notably, the *IDH1* R132 mutation was found to be more prevalent in CD than in UC [79], suggesting that disease-related differences in genetic alterations may influence carcinogenesis in CAC.

### 6.4. The Role of Gut Microbiota in CAC

Gut microbiota play a crucial role in maintaining gut homeostasis and regulating immune responses. Dysbiosis, an imbalance in microbial composition, is closely linked to IBD and CAC [95,96,97]. Advances in next-generation sequencing have deepened our understanding of how microbiota contribute to these conditions [98]. Factors related to the gut microbiota that influence CRC development include reactive oxygen species generation, toxin secretion, and biofilm formation, which can disrupt the mucosal barrier, induce intestinal inflammation, and damage enterocytes [99,100]. Various studies have identified bacterial species such as *Fusobacterium nucleatum* and *Escherichia coli* as being associated with CRC [101,102,103,104]. Interestingly, *F. nucleatum* is also enriched in the gut of patients with IBD [101]. Furthermore, the strains isolated from IBD tissues show significantly greater invasiveness than those from healthy tissues [102]. The virulence factors of *F. nucleatum*, such as FadA, bind to E-cadherin, activating the β-catenin pathway and inducing the expression of oncogenic factors (e.g., cyclin D1, NF-κB, and C-myc), promoting tumor cell growth and metastasis [101,105]. Specific *E. coli* strains harbor a non-ribosomal peptide synthase operon, hybrid polyketide (pks), which encodes colibactin, a genotoxin that induces DNA double-strand breaks. These breaks lead to mutations, chromosomal rearrangements, and cell cycle disruption, ultimately impairing normal cellular function [103,106,107]. Notably, pks+ *E. coli* strains are present in approximately 20% of healthy individuals, 40% of patients with IBD, and 67% of patients with CRC, suggesting a progressive association between microbial presence and carcinogenesis [104,108]. This finding implies that carcinogenic bacteria may contribute to intestinal mucosal damage in IBD, highlighting their potential role in IBD-associated carcinogenesis. On the other hand, probiotic therapy may help suppress tumorigenesis by modulating the gut microbiota and reducing inflammation.

### 6.5. Animal Models for CAC Research

Given the complex immunological and genetic mechanisms underlying CAC, investigations using animal models offer valuable insights into its pathophysiology and potential therapeutic strategies. These models help elucidate the interactions among inflammation, genetic alterations, and carcinogenesis, which are difficult to study in human patients. Various chemically induced and genetically engineered CAC animal models have been developed, with chemically induced models being particularly common. These models typically involve the administration of carcinogenic compounds, such as azoxymethane (AOM) or 1,2-dimethylhydrazine, followed by repeated cycles of pro-inflammatory agents, such as dextran sulfate sodium or 2,4,6-trinitrobenzenesulfonic acid [109,110]. AOM and 1,2-dimethylhydrazine are potent carcinogens that induce a broad spectrum of mutations in critical genes involved in intracellular signaling pathways. The combination of carcinogenic compounds and pro-inflammatory agents is particularly useful for studying the role of inflammatory cytokines and molecules in carcinogenesis because these models mimic the progression from inflammation to dysplasia and ultimately to cancer, similar to human CAC.

A murine model mimicking CAC demonstrated that cytokines and chemokines dependent on NF-κB, STAT3, such as IL-1β, IL-6, KC/CXCL1, and eotaxins, and p38 MAPK signaling pathways, were upregulated during the inflammation phase. As the disease progressed, cytokines and chemokines such as GM-CSF, G-CSF, SDF-1/CXCL12, and RANTES/CCR5, regulated by both Wnt/β-catenin and NF-κB signaling pathways, were particularly elevated in high-grade dysplasia and cancer in the murine CAC model [109]. These findings highlight dynamic shifts in activated pathways during the progression from inflammation to cancer. While chemically induced animal models are widely used, it is important to recognize their limitations in replicating the pathophysiological characteristics of human CAC. Notably, these models do not mimic *TP53* mutations, which are frequently observed in patients with IBD and are associated with flat lesions. Instead, AOM administration induces polypoid lesions, a characteristic feature of sCRC [111]. Recently, a novel murine CAC model has been developed in which the dominant–negative TGFβ receptor II is expressed in T cells (CD4-dnTGFβRII/AOM) [112]. In this model, macroscopically invisible flat adenocarcinoma lesions develop following AOM exposure, with loss of p53 indicated. Animal models provide valuable tools for investigating the complex pathogenesis of CAC, but it is essential to understand their characteristics and limitations because no model has been able to fully replicate human CAC to date.

### 6.6. Organoid Models for CAC Research

Given the limitations of animal models, there has been growing interest in organoid-based studies in recent years. Organoids provide a human-relevant platform for studying disease mechanisms and offer a more detailed understanding of pathophysiology, which animal models cannot fully replicate. A study using human colon organoids and whole-exome sequencing revealed that somatic mutations in multiple genes related to IL-17 signaling, including *NFKBIZ*, *ZC3H12A*, and *PIGR*, accumulated in the inflamed epithelium of patients with UC [113]. These genetic alterations in the colonic epithelium appear to be causally linked to the inflammatory cascade. Meanwhile, a study examining non-dysplastic UC colon and CAC showed that mutational profiles differed between these tissues, suggesting that specific mutations undergo positive selection in the affected tissues. The study also demonstrated that *NFKBIZ* mutations were highly prevalent in the epithelium of non-dysplastic UC colon but were rarely detected in CAC and sCRC [114]. This observation—that mutations accumulate due to chronic intestinal inflammation but decrease during carcinogenesis—initially appears contradictory but suggests that *NFKBIZ*-mutant cells are negatively selected during the CAC carcinogenesis process. Although the underlying mechanisms remain unclear, these findings imply that clonal selection occurs in UC and CAC tissues. Further investigations using organoid technology are expected to provide molecular insights into the carcinogenesis of CAC and help clarify these selection mechanisms.

## 7. Conclusions

This review provides an extensive and up-to-date overview of inflammation-associated carcinogenesis in IBD, covering clinical features, challenges, and molecular biological mechanisms. Despite advances in IBD treatment, the risk of cancer in long-term cases must still be considered. Understanding the endoscopic and pathological features of inflammation-associated neoplasia is essential, and optimizing surveillance remains a critical clinical challenge. Furthermore, although it has been reported that the carcinogenic mechanisms of inflammation-associated carcinogenesis differ from those of typical sporadic cancers, more evidence is needed to fully understand the many factors involved in these complex mechanisms. Research using animal models and organoids is actively being conducted. It is expected that a better understanding of these carcinogenic mechanisms will lead to new diagnostic and therapeutic approaches for inflammation-associated neoplasia in IBD.

## Figures and Tables

**Figure 1 cells-14-00567-f001:**
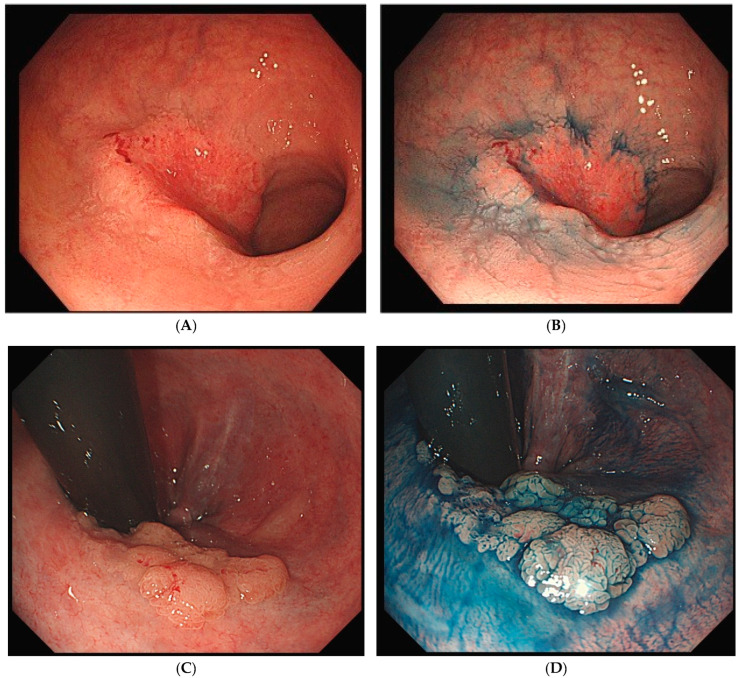
Endoscopic findings of UCAN. (**A**) Cancer (IIc) in the sigmoid colon with WLI. (**B**) Indigo carmine staining of (**A**). (**C**) Cancer (IIa) in the rectum with WLI. (**D**) Indigo carmine staining of (**C**). (**E**) Cancer (IIa + Is) in the rectum with WLI. (**F**) Indigo carmine staining of (**E**). (**G**) Cancer (IIb) in the rectum (rectosigmoid) with WLI. UCAN: UC-Associated Colorectal Neoplasia; WLI: white-light imaging.

**Figure 2 cells-14-00567-f002:**
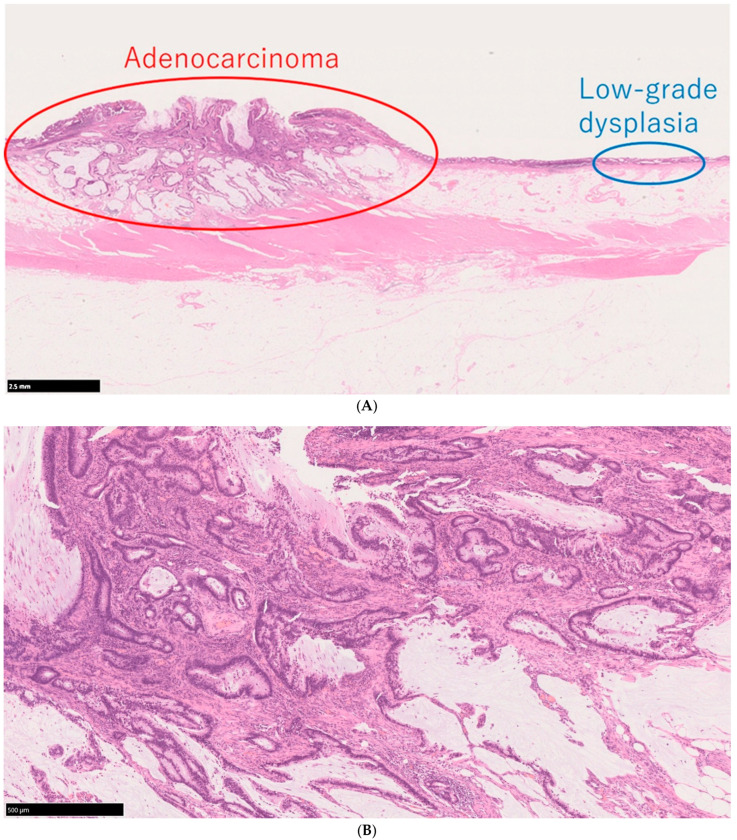
Histological findings of a case of UCAN. (**A**) A case of adenocarcinoma with low-grade dysplasia, HE staining. (**B**) Area of adenocarcinoma, HE staining. (**C**) p53 immunostaining of the same area. (**D**) Area of low-grade dysplasia, HE staining. (**E**) p53 immunostaining of the same area. Scale bar indicates 500 μm HE: hematoxylin and eosin; UCAN: UC-Associated Colorectal Neoplasia.

**Figure 3 cells-14-00567-f003:**
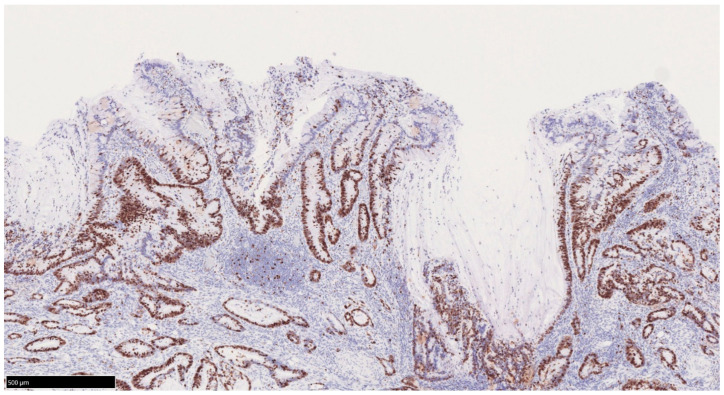
Bottom-up pattern of the proliferative zone in UCAN. Ki67 immunostaining. In UCAN, the proliferative zone is displaced to the deeper layer of the mucosa, and there is a tendency to show a so-called bottom-up pattern of superficial differentiation. UCAN: UC-Associated Colorectal Neoplasia.

**Figure 4 cells-14-00567-f004:**
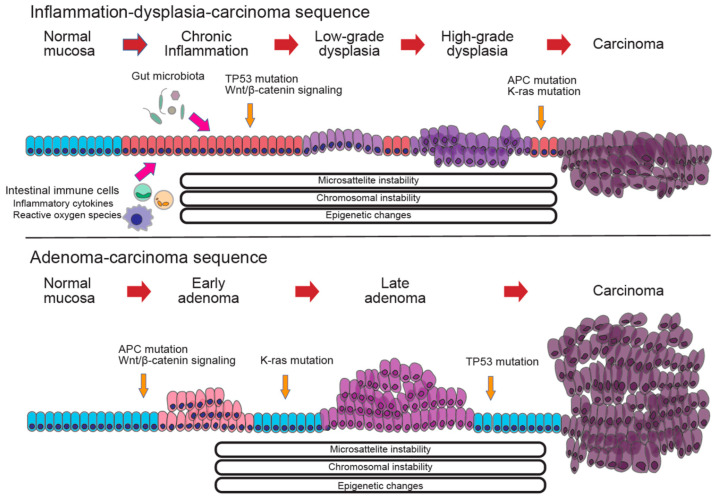
Comparison of the molecular mechanisms in colorectal cancer. (**Upper**) The molecular mechanism of the inflammation–dysplasia–carcinoma sequence, which is characterized by TP53 mutations in the early stages, and APC and K-ras mutations in the later stages. (**Lower**) The molecular mechanism of the adenoma–carcinoma sequence, which is characterized by APC and K-ras mutations in the early stages and TP53 mutations in the later stages.

## Data Availability

Not applicable.

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
