# Peer review of "Inflammation-Associated Carcinogenesis in Inflammatory Bowel Disease: Clinical Features and Molecular Mechanisms"

_cells, 2025, doi:10.3390/cells14080567_

Round 1
Reviewer 1 Report
Comments and Suggestions for Authors
Revision of the Paper
The paper “Inflammation-associated carcinogenesis in inflammatory bowel disease: clinical features and molecular mechanisms” by Hisamatsu T. et al. addresses a current and scientifically relevant topic. It covers all important aspects of inflammation-associated carcinogenesis in IBD, beginning with epidemiology and clinical risk factors, highlighting features of UCAN, surveillance colonoscopy, and concluding with important insights into the molecular mechanisms of CAC in IBD.
Section-specific concerns:
-
Epidemiology of inflammation-associated neoplasia in IBD (1):
Please provide a short conclusion or summary at the end of the paragraph. -
Patient-related clinical risk factors (2.1.):
Only PSC (primary sclerosing cholangitis) is mentioned as a patient-related risk factor for inflammation-associated carcinogenesis. A more detailed literature review should be conducted. -
Endoscopic and pathological features of UCAN (3) and surveillance colonoscopy (4):
Justify the chosen methodology (endoscopy and colonoscopy) in comparison to alternative approaches. -
Molecular mechanisms of colitis-associated cancer (CAC) in IBD (5):
While the previous sections are well-structured, this paragraph lacks a clear structure. It should be divided into subcategories such as:-
Wnt/ß-catenin pathway
-
Cytokines
-
CIN and DNA methylation
-
Gene mutations
-
General notes:
-
Highlight key findings and their practical significance.
-
Discuss potential limitations and suggest directions for future research.
-
Provide a clearer summary of the main findings.
Conclusion:
The paper provides a solid foundation but could be improved by implementing the suggested recommendations.
Author Response
Comment 1
Epidemiology of inflammation-associated neoplasia in IBD (1):
Please provide a short conclusion or summary at the end of the paragraph.
Response:
We have added a summary at the end of the paragraphs of UC- and CD-associated neoplasia and CD respectively as below:
UC-associated neoplasia
Overall, while the cumulative incidence of UCAN may not be as high as previously reported, careful monitoring for CRC remains necessary for patients with UC.
CD-associated neoplasia
Overall, patients with CD are considered to have a higher risk of gastrointestinal neoplasms compared to the general population. Inflamed lesions, not only in the colon but also in the anorectal area and small intestine, can increase the risk of developing cancer.
Comment 2
Patient-related clinical risk factors (2.1.):
Only PSC (primary sclerosing cholangitis) is mentioned as a patient-related risk factor for inflammation-associated carcinogenesis. A more detailed literature review should be conducted.
Response:
Several patient-related clinical risk factors are described in subsection 2.1. of the original manuscript. Meanwhile, crucial factors related to IBD, which need to be reviewed in detail, are mentioned in subsection 2.2. We believe this structure makes the key points less clear to the audience.
Therefore, we have reconsidered the structure of subsections 2.1 and 2.2., as both “patient-related clinical risk factors” and “disease-related risk factors” fall under the broader category of “patient-related risk factors.” This is subsection 3.1. in the revised manuscript (The numbering of sections is changed because we have added the Introduction section as section 1 in the revised manuscript). In addition to combining these subsections, we have added a sentence to improve the flow of the text.
Comment 3
Endoscopic and pathological features of UCAN (3) and surveillance colonoscopy (4):
Justify the chosen methodology (endoscopy and colonoscopy) in comparison to alternative approaches.
Response:
Today, several guidelines recommend an endoscopic approach for surveillance for UCAN, as we described in the original manuscript, citing some references. We have added this point to the beginning of subsection 4.1. in the revised manuscript to clarify the rationale for emphasizing endoscopy.
Comment 4
Molecular mechanisms of colitis-associated cancer (CAC) in IBD (5):
While the previous sections are well-structured, this paragraph lacks a clear structure. It should be divided into subcategories such as:
- Wnt/ß-catenin pathway
- Cytokines
- CIN and DNA methylation
- Gene mutations
Response:
We have reorganized this section into subsections, adding subtitles, following the reviewer’s suggestion.
Comment 5
General notes:
- Highlight key findings and their practical significance.
- Discuss potential limitations and suggest directions for future research.
- Provide a clearer summary of the main findings.
Comment 6
Conclusion:
The paper provides a solid foundation but could be improved by implementing the suggested recommendations.
Response:
The assistant editor pointed out that our original manuscript lacked Introduction and Conclusion sections. We have added these to the revised manuscript and believe this revision addresses the reviewer's comments.
Reviewer 2 Report
Comments and Suggestions for Authors
Major comments
This review mainly focuses on the clinical symptoms and molecular mechanisms of colorectal cancer related to inflammatory bowel disease. The entire article is well organized, and the content is introduced comprehensively. I only have the following suggestions:
Detail comments
- I suggest the author carefully check the author ranking and ranking tags (for example, the corresponding author does not seem to be accurately marked).
- I recommend combining the images in Figure 1 and Figure 2 for display.
- The impairment of intestinal barrier function may be an important cause of IBD, and it is recommended that the author supplement the role of intestinal barrier function in IBD related colon cancer.
Comments on the Quality of English Language
The author can check the overall English writing to improve the quality of the article.
Author Response
Comment 1
I suggest the author carefully check the author ranking and ranking tags (for example, the corresponding author does not seem to be accurately marked).
Response:
We thank the reviewer for the detailed check. We have used a designated pre-layout format to prepare the revised manuscript and carefully checked the author list.
Comment 2
I recommend combining the images in Figure 1 and Figure 2 for display.
Response:
We appreciate the reviewer’s suggestion. Since each figure contains many panels, it seems hard to combine all panels in one figure. However, we would like to request the editorial team to ensure that the two figures are presented close to each other when published.
Comment 3
The impairment of intestinal barrier function may be an important cause of IBD, and it is recommended that the author supplement the role of intestinal barrier function in IBD related colon cancer..
Response:
As the reviewer pointed out, impairment of intestinal barrier function is believed to be involved in IBD pathogenesis. However, based on our literature review, the role of intestinal barrier function in IBD-associated CRC remains unclear. We agree with the reviewer that this could be an interesting area of research, particularly in the context of IBD-associated cancer.